# Language Embedding Meets Dynamic Graph: A New Exploration for Neural Architecture Representation Learning

## Abstract

Neural Architecture Representation Learning aims to transform network models into feature representations for predicting network attributes, playing a crucial role in deploying and designing networks for real-world applications. Recently, inspired by the success of transformers, transformer-based models integrated with Graph Neural Networks (GNNs) have achieved significant progress in representation learning. However, current methods still have some limitations. First, existing methods overlook hardware attribute information, which conflicts with the current trend of diversified deep learning hardware and limits the practical applicability of models. Second, current encoding approaches rely on static adjacency matrices to represent topological structures, failing to capture the structural differences between computational nodes, which ultimately compromises encoding effectiveness. In this paper, we introduce LeDG-Former, an innovative framework that addresses these limitations through the synergistic integration of language-based semantic embedding and dynamic graph representation learning. Specifically, inspired by large language models (LLMs), we propose a language embedding framework where both neural architectures and hardware platform specifications are projected into a unified semantic space through tokenization and LLM processing, enabling zero-shot prediction across different hardware platforms for the first time. Then, we propose a dynamic graph-based transformer for modeling neural architectures, resulting in improved neural architecture modeling performance. On the NNLQP benchmark, LeDG-Former surpasses previous methods, establishing a new SOTA while demonstrating the first successful cross-hardware latency prediction capability. Furthermore, our framework achieves superior performance on the cell-structured NAS-Bench-101 and NAS-Bench-201 datasets. The source code will be released publicly.

## 1 Introduction

With the rapid development of deep learning technology, an increasing number of various neural networks are designed and deployed in real-world applications (Chen et al., 2018; Dudziak et al., 2020; Zhang et al., 2021; Liu et al., 2022; Baylor et al., 2017). This progression has facilitated the practical adoption of technologies, but simultaneously increased the workload for model deployment and development. To address this challenge, researchers have proposed neural architecture representation learning, leveraging deep learning techniques themselves to accelerate both model deployment and novel model development (Wen et al., 2020; Ning et al., 2020; Yi et al., 2023a;b). The purpose of neural architecture representation learning is to encode network structures into feature vectors, enabling subsequent attribute prediction based on these representations. The encoding process requires careful consideration of both operational node attributes and topological structure information of the network (Wen et al., 2020; Ning et al., 2020; Yi et al., 2023a;b). Neural architecture representation learning supports various downstream tasks, such as performance prediction, hardware deployment optimization, and Neural Architecture Search (NAS) (Luo et al., 2018; Cai et al., 2019; Luo et al., 2020; Xu et al., 2021; Chen et al., 2021; Qin et al., 2022).

In neural architecture representation learning, neural architectures are naturally expressed as Directed Acyclic Graphs (DAGs) (Cai et al., 2018; Zela et al., 2019; Li et al., 2020; Dong et al., 2022; Luo et al.,

2023), where nodes correspond to computational operations and edges represent data flow between them. With the emergence of Graph Neural Networks (GNNs) and their demonstrated effectiveness in related work, early approaches commonly relied on GNNs, leveraging graph convolution to capture adjacency relationships between nodes for explicit modeling of these DAGs, thereby achieving preliminary representations of neural network structures. Representative methods such as Peephole, BRP-NAS, GATES, BANANAS, and NNLP (Deng et al., 2017; Dudziak et al., 2020; Ning et al., 2020; White et al., 2021; Liu et al., 2022) adopted this strategy. However, due to the inherent locality of GNNs' aggregation mechanisms, these methods exhibit limitations in representing complex cross-layer topological information (Kipf & Welling, 2016; Veličković et al., 2017). To overcome these limitations, Transformer architectures have gradually been introduced into neural architecture representation learning. By leveraging their powerful global attention mechanisms, they improve the quality of structural representation. Representative methods like TNASP and NAR-Former (Lu et al., 2021; Yi et al., 2023a) utilize self-attention mechanisms to capture global semantic associations between nodes, significantly enhancing model performance. The Transformer-based representation learning method benefits from the flexibility of self-attention mechanisms, demonstrating remarkable effectiveness on cell-structured datasets such as NAS-Bench-101 (Ying et al., 2019) and NAS-Bench-201 (Dong & Yang, 2020). However, the global receptive field characteristic of Transformers makes them particularly sensitive when encoding long sequences, resulting in relatively weaker generalization capabilities (Yi et al., 2023a).

Recent research has attempted to introduce graph structure enhancement mechanisms within Transformer frameworks. For instance, Graphormer(Ying et al., 2021) and GraphTrans(Wu et al., 2021) both inject graph-structured attention masks into Transformers to simulate message passing, enabling structure-aware encoding that benefits architecture performance prediction. NAR-Former V2 (Yi et al., 2023b) proposed a position-aware graph embedding technique that explicitly integrates adjacency relationships into the attention mechanism, thereby improving prediction accuracy. GNN-Enhanced Transformer(Xiang et al., 2024) proposes a unified framework that combines GNN-based local topology encoding with Transformer-based global modeling, achieving improved performance prediction through joint structural reasoning. NN-Former (Xu et al., 2025) incorporated forward, backward, and same-layer adjacency information into attention calculations to achieve richer topological representations, enhancing both accuracy and generalization. Such Transformer frameworks embedded with GNN mechanisms have demonstrated strong capabilities.

Although Transformer-GNN hybrid methods for neural architecture representation learning inherit the flexibility of Transformers and the topological encoding strengths of GNNs, achieving significant performance improvements, these approaches still face several limitations. First, existing methods primarily focus on encoding the network architecture itself while neglecting hardware attributes. However, inference efficiency post-deployment is highly dependent on hardware characteristics, and this omission significantly limits the applicability of representation learning approaches. Moreover, with the proliferation of specialized hardware for AI models, this limitation will become increasingly impactful. Second, current GNN-based approaches predominantly rely on static adjacency matrices to capture topological information, failing to account for positional variations among nodes and their distinct neighborhood attention patterns. This oversight constrains the modeling capacity for topological structure representation.

In this paper, inspired by LLM, we conduct a new exploration and combining language embedding and dynamic graph to address these limitations. Our major contributions can be summarized as: 1) The innovative use of LLMs' powerful language encoding capabilities to jointly map hardware specifications and network architecture details into a unified semantic space. This enables hardware-software co-optimized representation learning for neural networks. Unlike prior methods limited to single-hardware optimization, our approach facilitates zero-shot cross-hardware attribute prediction; 2) To ensure high-quality encoding, we conducted a thorough analysis of LLM encoding characteristics and designed specialized language templates. Leveraging the LLM's capabilities, we serialized both network structures and hardware information. Furthermore, we introduce dynamic graph self-attention, a novel mechanism that improves flexibility in capturing topological relationships across nodes, thereby enhancing representation effectiveness.

## 2 RELATED WORKS

### 2.1 GNN FOR REPRESENTATION LEARNING

Neural Architecture Representation Learning has emerged as a vital tool for predicting model attributes such as accuracy, latency, and energy consumption, especially under cross-platform deployment scenarios. A key insight in this field is that neural architectures can be naturally represented as Directed Acyclic Graphs (DAGs), where nodes denote computational operations and edges represent data flows. Early approaches, such as Peephole (Deng et al., 2017)and BRP-NAS (White et al., 2021) utilized handcrafted global descriptors or structural metrics derived from DAGs, such as operation counts or edge lists, to encode architectural features. However, these static encodings failed to capture the expressive structural nuances of complex models.

To better model DAG structures, Graph Neural Networks were introduced. Methods like GATES (Ning et al., 2020), arch2vec (Yan et al., 2020) and TA-GATES (Ning et al., 2022) use adjacency matrices and node-level attributes to perform message passing over the DAG, enabling localized structural representation and improved generalization to unseen architectures. These models successfully capture some topological semantics through fixed edge types, but are fundamentally limited by the locality and rigidity of their aggregation functions. In particular, they struggle to model long-range dependencies or dynamically adapt relational attention across diverse network structures (Scarselli et al., 2008; Hamilton et al., 2017; Xu et al., 2018; Dwivedi et al., 2023). This structural rigidity and limited expressiveness of GNNs highlight the need for more flexible, context-aware models. Consequently, research has shifted toward attention-based alternatives, particularly Transformer architectures, which are better suited for learning long-range interactions in heterogeneous structures.

### 2.2 TRANSFORMER FOR REPRESENTATION LEARNING

In response to the limitations of GNN-based models, Transformer architectures have been adopted for Neural Architecture Representation Learning due to their ability to capture long-range dependencies and model flexible interaction patterns. Initial Transformer-based methods such as TNASP (Lu et al., 2021) and NAR-Former (Yi et al., 2023a) represent architectures as sequences of operation or connection tokens, applying self-attention to learn global semantic relationships. However, these sequence-based representations lack explicit structural bias, making them sensitive to minor topological variations and insufficient for capturing the inherent graph properties of architectures.

To incorporate structural information more directly, hybrid approaches have emerged. NAR-Former V2 (Yi et al., 2023b) introduces topology-aware token connections, embedding adjacency patterns into the attention mechanism. NN-Former (Xu et al., 2025) goes further by disentangling multiple structural relations, such as hierarchical, sibling, and descendant dependencies, and embedding them through graph-aware attention kernels within a Transformer encoder. These improvements enhance the model's capacity to reason over complex DAGs and achieve state-of-the-art results. However, both methods still rely on fixed structural priors, where adjacency relations are statically defined and shared between architectures. This overlooks the dynamic relevance of different topological views for different network instances.

### 2.3 EMBEDDING STRATEGY FOR REPRESENTATION LEARNING

The embedding strategy plays a pivotal role in determining the quality and generalization of neural architecture representations. Earlier approaches primarily focused on embedding the structural aspects of neural architectures, such as node operations and topological patterns (Zoph & Le, 2016; Liu et al., 2018; Deng et al., 2017; Dudziak et al., 2020; Hamilton et al., 2017). These methods often relied on simple vectorization techniques that lacked semantic richness, limiting the amount of meaningful information captured from the architecture.

With the growing adoption of Transformers, such as TNASP (Lu et al., 2021), NAR-Former (Yi et al., 2023a) and Autogt (Zhang et al., 2023) introduced position-aware embeddings that tokenize architectural structures for attention-based modeling. More recent models, including NAR-Former V2 and NN-Former, further incorporate static attributes of neural networks by embedding them separately alongside the structure, such as flops, depth, and batch size. These methods are specifically designed

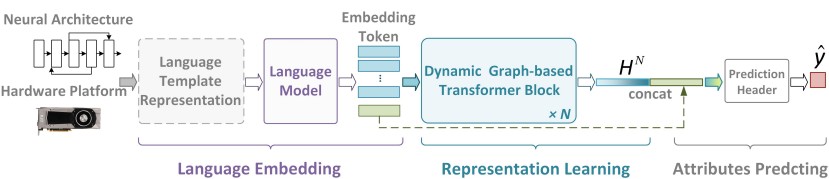

Figure 1: Overview of the proposed LeDG-Former.

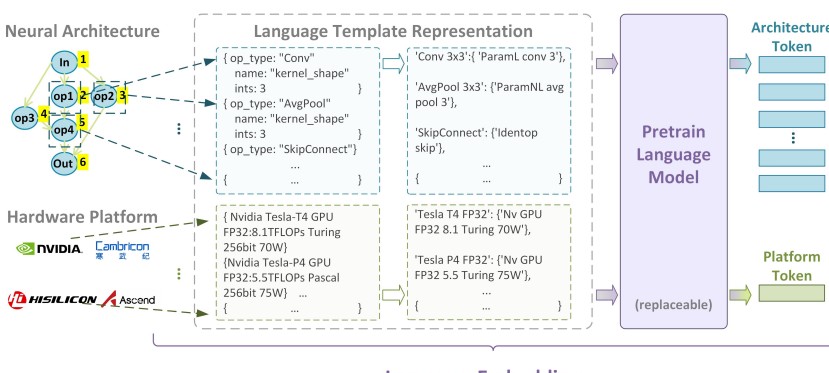

Figure 2: Illustration of the proposed language embedding

encoding approaches tailored for network structure representation, exhibiting poor extensibility. For instance, they would fail when encountering unseen node types or novel hyperparameters. Moreover, these encoding schemes primarily focus on the network architecture itself while neglecting hardware-related information.

## 3 METHODS

The final framework of LeDG-Former is shown in Fig.1, which consists of two key stages: a language embedding stage using a pre-trained language model and a representation learning stage employing dynamic graph-aware self-attention. In the language embedding stage, we systematically encode both model architecture information and hardware platform specifications through carefully designed linguistic templates, then transform them into feature tokens using a pre-trained LLM. These embedding tokens serve as input to our dynamic graph-aware self-attention mechanism that adaptively models node-level dependencies in the computational graph while capturing cross-modal interactions between hardware and architecture features. The resulting network representation token is concatenated with the hardware platform's language embedding for final attribute prediction. Next, we will provide a detailed explanation of these two stages.

### 3.1 LANGUAGE EMBEDDING

The language embedding module is designed to encode both neural architecture information and hardware specifications into feature vectors within a unified representation space. As shown in Fig.2, this paper adapts the tokenizer from pretrained language models (LLMs) to achieve this joint mapping. For neural architectures, the network architecture is first represented as a directed acyclic graph (DAG) following the node sequence. For each node in the graph, we extract its information according to predefined language template. These structured descriptions are then fed into the LLM and compressed into a unified feature vector representation. A similar process is adopted in modeling hardware platform information. Different language templates are designed for modeling neural architecture information and hardware platform information:

- When designing language templates for neural architectures, our primary consideration is to ensure accurate and concise descriptions of operations and their attributes so that the

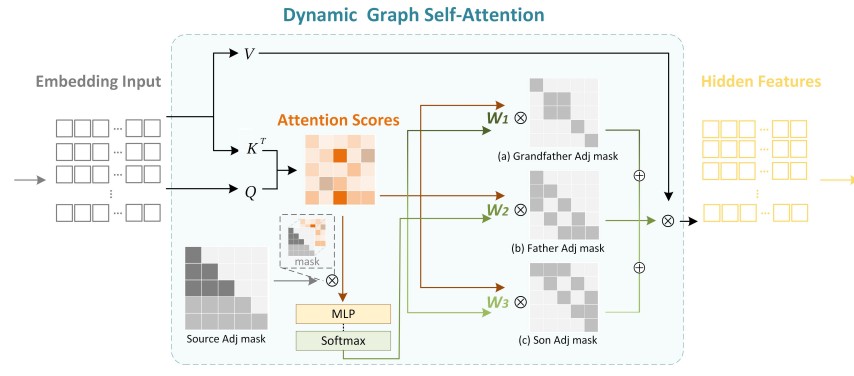

Figure 3: Diagram of the proposed Dynamic Graph Self-Attention (DGSA).

embedded information remains faithful. First, we observe that different types of operations may affect the target prediction differently. Therefore, classifying the operation types during standardization helps preserve this information. Meanwhile, for operation-specific attributes, such as the kernel size of a convolution operation, we represent them using concise numerical tokens to prevent such attributes from being overwhelmed by surrounding context in the language embedding process. For example, "Conv 3×3" is extracted and described using the template as "ParamL Conv 3", where "ParamL" serves as a template indicator for "operations with learnable parameters".

- For hardware platform information, we focus on platform attributes that are impactful for latency prediction. We prioritize information such as computational throughput and power consumption under different inference precisions, which directly influence model latency. Furthermore, to support cross-platform generalization tasks, it is also important to include platform type and architectural-level descriptions in the template. For example, the Nvidia Tesla T4 under FP32 precision is described using the template as "Nv GPU FP32 8.1 Turing 70W".

The language embedding for the node $i$ is generated by:

$$f_{node_i} = \text{LLM}(\text{Tokenizer}(\text{T}_{\text{arch}}(info_i))), \tag{1}$$

where $f_{node_i}$ is the language embedding, and $\text{T}_{\text{arch}}$ represents language template for neural architecture. $info_i$ denotes the information of the $i-$th node. The LLM adopted here is not limited to a specific one, this paper adopts BERT. The language embedding for platform is calculated as:

$$f_{plat} = \text{LLM}(\text{Tokenizer}(\text{T}_{\text{plat}}(info_{plat}))), \tag{2}$$

where $info_{plat}$ is the platform information. Both LLM and $Tokenizer$ adopted here are same with that adopted in Equation (1), which ensures the neural architecture information and platform information are projected in the same space. For a neural architecture with $n$ nodes, the output of language embedding stage is $[f_{node_1}, f_{node_2}, \ldots, f_{node_n}, f_{plat}]$.

## 3.2 DYNAMIC GRAPH SELF-ATTENTION

Following the research line of combining transformer and GNN for representation learning (Yi et al., 2023b; Xu et al., 2025), we propose Dynamic Graph Self-Attention (DGSA) and employ it to replace the standard self-attention mechanism in Transformers. Unlike prior works that rely on static adjacency matrices to model topological structures, the proposed DGSA dynamically aggregates multi-scale topological information by adaptively retrieving relevant connectivity patterns from three hierarchical contexts: (1) grandfather nodes (two-hop predecessors), (2) father nodes (direct predecessors), and (3) son nodes (direct successors), as shown in Fig.3. This design facilitates adaptive topology-aware representation learning, leading to consistent performance gains, as verified in ablation study part.

Specifically, this process contains two steps. The dynamic weights are computed by incorporating information from predecessor nodes, with the formula:

$$f^w_{node_i} = \text{Softmax}(q_i \cdot (k_1, k_2, \ldots, k_i))(v_1, v_2, \ldots, v_i), \tag{3}$$

$$W_1, W_2, W_3 = \text{Softmax}(\text{MLP}(f^w_{node_i})), \tag{4}$$

where $q_i = W^{dw}_q f_{node_i}$, $k_i = W^{dw}_k f_{node_i}$, $v_i = W^{dw}_v f_{node_i}$. MLP represents a fully connected layer with three output nodes. The final representation is calculated with formula:

$$f^r_{node_i} = \sum_{i=1}^{3} W_i \cdot X_i, \tag{5}$$

$$X_1 = \sigma\left(\left(QK^\top \circ (I + \mathbf{M}_{\text{Grandfather}})\right)/\sqrt{h}\right) V, \tag{6}$$

$$X_2 = \sigma\left(\left(QK^\top \circ (I + \mathbf{M}_{\text{Father}})\right)/\sqrt{h}\right) V, \tag{7}$$

$$X_3 = \sigma\left(\left(QK^\top \circ (I + \mathbf{M}_{\text{Son}})\right)/\sqrt{h}\right) V, \tag{8}$$

where $f^r_{node_i}$ is the representation learning feature of the $i-$th node. $Q = FW^Q$, $K = FW^K$, $V = FW^V$ denote the query, key, value. $F = [f_{node_1}, f_{node_2}, \ldots, f_{node_n}]$ is the language embedding result for neural architecture. $I$ is identity matrix, which ensures that each node can also attend to itself when computing adjacency-based attention. $M_{Grandfather}$, $M_{Father}$, $M_{Son}$ denote the masks derived from the adjacency matrices corresponding to grandfather, father, and son nodes. The derivation of these three masks is as follows: Let the binarized adjacency matrix corresponding to son nodes be denoted as $A$ ($M_{Son} = A$). Then $M_{Father} = Bi(A^T)$, and $M_{Grandfather} = Bi(A^T A^T)$, where $Bi$ is the binarization function.

## 4 EXPERIMENTS

In this section, we conduct experiments on three widely used neural architecture datasets: NNLQP (Liu et al., 2022), NAS-Bench-101 (Ying et al., 2019), and NAS-Bench-201 (Dong & Yang, 2020), to evaluate the effectiveness of our proposed framework. A series of ablation studies in Section 4.3 further validate the effectiveness of our design choices. Further experiments and implementation details related to training are included in the supplementary material.

### 4.1 LATENCY PREDICTION ON NNLQP

In this section, we perform latency prediction on the "unseen" datasets of the NNLQP to evaluate the effectiveness and generalization capability of our proposed framework. This dataset offers a diverse and comprehensive benchmark, comprising 20,000 deep learning networks across 10 distinct architecture types (2,000 samples per type). We compare our method against eight representative approaches, spanning from early linear regression-based prediction methods to recent representation learning frameworks.

We consider two different experiments. The first is a practically meaningful setting, where the target network type to be predicted does not appear in the training process. This experiment is divided into ten groups, where in each group, all samples of one network type are used as the test set, while samples of the remaining nine network types are used as the training set. As shown in Table 1, our method achieves the best performance in terms of both average MAPE and Acc(10%), averaged over 10 repeated experimental runs. Compared to the second-best method, NN-Former, our approach improves the average Acc(10%) by 2.29% and reduces average MAPE by 0.66. These results demonstrate that our proposed self-attention mechanism with dynamic adjacency awareness enables each node to attend to more appropriate topological information, resulting in more accurate neural architecture representations.

In the second experiment, the training and testing sets are drawn from the same network types distribution, as shown in Table 2. We construct the training set using the first 1,800 samples from each of the ten network types, and the remaining 2,000 networks are used as the test set. When testing on all network types test samples, our method achieves a highest average Acc(10%) and

Table 1: Out of domain latency prediction on NNLQP. "Test Model = AlexNet" means that only AlexNet models are used for testing, and the other 9 model families are used for training. The best results refer to the lowest MAPE and corresponding Acc (10%) in 10 repeated experiments. TPU (Kaufman et al., 2021).

| Metric | Test Domain | FLOPs | FLOPs+MAC | nn-Meter | TPU | BRP-NAS | NNLP (avg/best) | NAR-FormerV2 (avg/best) | NN-Former (avg/best) | Ours (avg/best) |
|---|---|---|---|---|---|---|---|---|---|---|
| MAPE ↓ | AlexNet | 44.65 | 15.45 | 7.20 | 10.55 | 31.68 | **10.64 / 9.71** | 24.28 / 18.29 | 11.47 / 11.17 | 10.92 / 10.88 |
| | EfficientNet | 58.36 | 53.96 | 18.93 | 16.74 | 51.97 | 21.46 / 18.72 | 13.20 / 11.37 | 5.13 / 4.81 | **4.61 / 4.54** |
| | GoogleNet | 30.76 | 32.54 | 11.71 | 8.10 | 25.48 | 13.28 / 10.90 | 6.61 / 6.15 | 6.74 / 6.65 | **5.50 / 5.39** |
| | MnasNet | 40.31 | 35.96 | 10.69 | 11.61 | 17.26 | 12.07 / 10.86 | 7.16 / 5.93 | **2.71 / 2.54** | 3.31 / 3.01 |
| | MobileNetV2 | 37.42 | 35.27 | 6.43 | 12.68 | 20.42 | 8.87 / 7.34 | 6.73 / 5.65 | **4.17 / 3.66** | 4.29 / 4.06 |
| | MobileNetV3 | 64.64 | 57.13 | 35.27 | 9.97 | 58.13 | 14.57 / 13.17 | 9.06 / 8.72 | 9.07 / 9.03 | **8.30 / 8.06** |
| | NasBench201 | 80.41 | 33.52 | 9.57 | 58.94 | 13.28 | 9.60 / 8.19 | 9.21 / 7.89 | **7.93 / 7.71** | 8.33 / 7.84 |
| | ResNet | 21.18 | 18.91 | 15.58 | 20.05 | 15.84 | 7.54 / 7.12 | 6.80 / 6.44 | 7.49 / 7.38 | **6.71 / 6.66** |
| | SqueezeNet | 29.89 | 23.19 | 18.69 | 24.60 | 42.55 | 9.84 / 9.52 | 7.08 / 6.56 | 9.08 / 7.05 | **5.85 / 5.85** |
| | VGG | 69.34 | 66.63 | 19.47 | 38.73 | 30.95 | **7.60 / 7.17** | 15.40 / 14.26 | 20.12 / 19.64 | 19.45 / 17.86 |
| | Average | 47.70 | 37.26 | 15.35 | 21.20 | 30.76 | 11.55 / 10.27 | 10.55 / 9.13 | 8.39 / 7.96 | **7.73 / 7.41** |
| Acc(10%) ↑ | AlexNet | 6.55 | 40.50 | **75.45** | 57.10 | 15.20 | 59.07 / 64.40 | 24.65 / 28.60 | 56.08 / 57.10 | 59.15 / 59.65 |
| | EfficientNet | 0.05 | 0.05 | 23.40 | 17.00 | 0.10 | 25.37 / 28.80 | 44.01 / 50.20 | 90.85 / 90.90 | **91.85 / 92.25** |
| | GoogleNet | 12.75 | 9.80 | 47.40 | 69.00 | 12.55 | 36.30 / 48.75 | 80.10 / 83.35 | 80.43 / 83.40 | **86.52 / 87.20** |
| | MnasNet | 6.20 | 9.80 | 60.95 | 44.65 | 34.30 | 55.89 / 61.25 | 73.46 / 81.60 | **98.65 / 98.70** | 97.45 / 98.40 |
| | MobileNetV2 | 6.90 | 8.05 | 80.75 | 33.95 | 29.05 | 63.03 / 72.50 | 78.45 / 83.80 | **94.90 / 96.85** | 92.65 / 95.05 |
| | MobileNetV3 | 0.05 | 0.05 | 23.45 | 64.25 | 13.85 | 43.26 / 49.65 | 68.43 / 70.50 | 74.18 / 74.30 | **74.46 / 75.85** |
| | NasBench201 | 0.00 | 10.55 | 60.65 | 2.15 | 43.45 | 60.70 / 70.60 | 63.13 / 71.70 | 69.78 / 71.10 | **69.90 / 72.70** |
| | ResNet | 26.50 | 29.80 | 39.45 | 27.30 | 39.80 | 72.88 / 76.40 | 77.24 / 79.70 | 70.83 / 71.55 | **77.93 / 78.75** |
| | SqueezeNet | 16.10 | 21.35 | 36.20 | 25.65 | 11.85 | 58.69 / 60.40 | 75.01 / 79.25 | 77.85 / 80.95 | **83.10 / 84.50** |
| | VGG | 4.80 | 2.10 | 26.50 | 2.60 | 13.20 | **71.04 / 73.75** | 45.21 / 45.30 | 29.40 / 29.85 | 33.12 / 36.27 |
| | Average | 7.99 | 13.20 | 47.42 | 34.40 | 21.34 | 54.62 / 60.65 | 62.70 / 67.40 | 74.31 / 75.47 | **76.60 / 78.06** |

Table 2: In domain latency prediction on NNLQP. Training and testing on the same distribution.

| Test Domain | MAPE ↓ | | | Acc(10%) ↑ | | |
|---|---|---|---|---|---|---|
| | NNLP (avg/best) | NN-Former (avg/best) | Ours (avg/best) | NNLP (avg/best) | NN-Former (avg/best) | Ours (avg/best) |
| AlexNet | 6.37 / 6.21 | **4.69 / 4.61** | 5.26 / 4.99 | 81.75 / 84.50 | **90.50 / 91.00** | 90.10 / 90.50 |
| EfficientNet | 3.04 / 2.82 | **2.31 / 2.21** | 2.61 / 2.50 | 98.00 / 97.00 | 99.00 / 100.0 | **99.60 / 100.00** |
| GoogleNet | 4.18 / 4.12 | 3.48 / 3.39 | 3.29 / 3.22 | 93.70 / 93.50 | 97.15 / 97.50 | **97.40 / 98.00** |
| MnasNet | 2.60 / 2.46 | 1.52 / 1.48 | **1.48 / 1.42** | 97.70 / 98.50 | 99.50 / 100.0 | **100.00 / 100.00** |
| MobileNetV2 | 2.47 / 2.37 | 1.54 / 1.50 | **1.43 / 1.34** | 99.30 / 99.50 | 99.60 / 100.0 | **100.00 / 100.00** |
| MobileNetV3 | 3.50 / 3.43 | 3.17 / 2.99 | **2.83 / 2.78** | 95.35 / 96.00 | 96.50 / 97.00 | **98.10 / 98.50** |
| NasBench201 | 1.46 / 1.31 | **1.11 / 0.96** | 1.16 / 1.11 | 100.0 / 100.0 | 100.0 / 100.0 | **100.00 / 100.00** |
| SqueezeNet | 4.03 / 3.97 | 3.09 / 3.08 | **2.58 / 2.49** | 93.25 / 93.00 | 97.70 / 98.00 | **99.60 / 100.00** |
| VGG | 3.73 / 3.63 | **2.94 / 2.89** | 3.06 / 2.99 | 95.25 / 96.50 | 95.80 / 96.50 | **96.50 / 97.50** |
| ResNet | 3.34 / 3.25 | **2.66 / 2.47** | 2.95 / 2.86 | 98.40 / 98.50 | **99.45 / 99.50** | 98.40 / 99.50 |
| All | 3.47 / 3.44 | 2.85 / 2.65 | **2.64 / 2.54** | 95.25 / 95.50 | 97.45 / 97.85 | **97.94 / 98.15** |

the best average MAPE. When testing on each network type individually, our method consistently outperforms NN-Former on all model types, except for the AlexNet and ResNet families, where the performance is comparable. These results further validate the effectiveness of our proposed self-attention mechanism with dynamic adjacency awareness, which enables more precise modeling of topological relationships among nodes.

## 4.2 HARDWARE AWARE ZERO-SHOT

In the zero-shot latency prediction across hardware platforms experiment, we perform an in-depth reorganization and mining of the data in the NNLQP "multi_platform" datasets, from which we extract latency samples under four inference configurations across two hardware platforms (Nvidia Tesla P4 and T4) and two numerical precisions (FP32 and INT8). The reorganized datasets contain 5,194 samples in total, including 1,416 and 1,075 samples for P4 under FP32 and INT8 respectively, and 1,150 and 1,553 samples for T4 under FP32 and INT8 respectively. Due to the relatively small number of samples and observable distributional discrepancies across different configurations, we adopt a pretrain-finetune strategy. First, we pretrain on the NNLQP "unseen" datasets (using the same datasets as in Section 4.1), and then finetune it on latency samples from T4 or P4 under different precision, in order to enable latency prediction on previously unseen hardware-precision combinations. To evaluate the effectiveness of our approach, we compare it against three baseline methods: linear predictors using FLOPs, FLOPs+MACs, and the NN-Former framework (Xu et al., 2025). The linear models serve as traditional baselines commonly used for cross-hardware latency estimation, while NN-Former represents the current state-of-the-art in learning-based latency prediction. Consistent with previous

Table 3: Zero-shot latency prediction on reorganized NNLQP "multi_platform" datasets. **Nvidia Tesla P4→Nvidia Tesla T4** means using latency sample on Tesla P4 for finetune, and zero-shot prediction on Tesla P4 sample.

| Metric | Test Domain | Nvidia Tesla P4→Nvidia Tesla T4 | | | | Nvidia Tesla T4→Nvidia Tesla P4 | | | |
|---|---|---|---|---|---|---|---|---|---|
| | | FLOPs | FLOPs+MAC | NN-Former | Ours | FLOPs | FLOPs+MAC | NN-Former | Ours |
| MAPE↓ | AlexNet | 326.99 | 431.72 | 32.66 | 97.95 | 350.29 | 552.4 | 92.49 | 79.17 |
| | EfficientNet | 49.64 | 28.92 | 34.81 | 34.96 | 43.83 | 25.02 | 37.71 | 19.12 |
| | GoogleNet | 27.25 | 37.53 | 68.69 | 20.54 | 50.13 | 28.39 | 46.92 | 19.09 |
| | MnasNet | 30.76 | 21.42 | 58.39 | 18.3 | 24.47 | 20.2 | 49.87 | 25.31 |
| | MobileNetV2 | 37.61 | 32.52 | 53.30 | 17.51 | 20.96 | 17.86 | 51.93 | 29.56 |
| | MobileNetV3 | 85.08 | 63.58 | 14.46 | 77.84 | 57.05 | 35.23 | 24.83 | 15.78 |
| | ResNet | 59.92 | 28.14 | 75.73 | 16.67 | 273.90 | 180.92 | 41.69 | 25.39 |
| | SqueezeNet | 41.77 | 25.86 | 71.51 | 29.81 | 166.98 | 91.5 | 46.85 | 40.11 |
| | VGG | 27.20 | 32.42 | 80.04 | 20.05 | 72.11 | 102.96 | 62.83 | 73.47 |
| | Average | 52.58 | 39.24 | 54.13 | **19.06** | 115.22 | 75.03 | 40.82 | **18.43** |
| Acc(10%)↑ | AlexNet | 0.00 | 0.00 | 0.00 | 0.00 | 0.00 | 0.00 | 0.00 | 9.67 |
| | EfficientNet | 0.00 | 5.03 | 5.02 | 14.57 | 0.55 | 16.39 | 0.55 | 30.21 |
| | GoogleNet | 14.00 | 3.00 | 0.00 | 30.10 | 5.05 | 20.71 | 0.00 | 29.00 |
| | MnasNet | 8.16 | 22.45 | 0.00 | 38.78 | 44.12 | 41.18 | 0.00 | 19.05 |
| | MobileNetV2 | 6.12 | 2.04 | 2.04 | 34.69 | 31.82 | 40.91 | 0.00 | 10.88 |
| | MobileNetV3 | 8.50 | 12.5 | 40 | 3.50 | 9.74 | 14.87 | 7.69 | 49.50 |
| | ResNet | 15.00 | 26.00 | 0.00 | 34.00 | 0.00 | 0.00 | 4.50 | 12.00 |
| | SqueezeNet | 17.00 | 19.00 | 0.00 | 9.50 | 0.00 | 0.00 | 0.51 | 4.50 |
| | VGG | 9.30 | 20.93 | 0.00 | 27.91 | 0.00 | 0.00 | 0.00 | 0.00 |
| | Average | 10.43 | 13.22 | 7.91 | **34.62** | 4.84 | 11.44 | 3.81 | **39.07** |

Table 4: Ablation study on NNLQP SqueezeNet family. Validate the effectiveness of DGSA and investigating the impact of embedding strategies with pretrained language models.

| Row | Embedding Strategies | | Graph Attention | | | | Metric | |
|---|---|---|---|---|---|---|---|---|
| | Position Embedding | Language model Embedding | ASMA | Global Attention | DGSA w/o Dynamic Graph Attention | DGSA | MAPE↓ | Acc(10%)↑ |
| 1 (Baseline: NN-Former) | ✓ | - | ✓ | - | - | - | 9.08 | 77.85 |
| 2 | - | ✓ | - | ✓ | - | - | 6.70 | 76.50 |
| 3 | - | ✓ | - | - | ✓ | - | 6.48 | 78.05 |
| 4 | ✓ | - | - | - | - | ✓ | 6.09 | 81.10 |
| 5 | - | ✓ (w/o pretrain) | - | - | - | - | 8.25 | 66.45 |
| 6 (Ours) | - | ✓ | - | - | - | ✓ | **5.85** | **83.10** |

studies (Yi et al., 2023a;b), we employ two standard evaluation metrics for latency prediction: Mean Absolute Percentage Error (MAPE) and Error Bound Accuracy (Acc(10%)). Specifically, Acc(10%) denotes the percentage of predictions with a relative error less than 10%.

As shown in Table 3, our method demonstrates promising and robust performance under two distinct zero-shot latency prediction settings across hardware platforms, performing favorably compared to conventional baselines. In the Nvidia Tesla P4→T4 experiment, we finetune on latency samples from P4 (under FP32 and INT8) and T4 (under INT8), and perform zero-shot prediction on previously unseen T4 FP32 samples. Our method achieves the best performance, with an Acc(10%) of 36.62% and a MAPE of 19.06. In the T4→ P4 setting, Our method again achieves the best performance, with 39.07% Acc(10%) and a MAPE of 18.43. These results outperform all baselines and highlight the effectiveness of incorporating hardware-aware modeling. In particular, the NN-Former results further support our observation in Section 1 that prior methods tend to overlook hardware attributes, which limits their generalization ability in cross-platform latency prediction tasks.

Overall, LeDG-Former integrates hardware-awareness via language-based embedding and exhibits strong generalization across diverse hardware platforms. As shown in the two experiments in Table 3, our method enables zero-shot latency prediction not only across different hardware configurations, but also across numerical precisions, from high-precision (FP32) to low-precision (INT8) settings on the same device, which is a critical feature for real-world model deployment scenarios that demand adaptability and efficiency.

## 4.3 ABLATION STUDIES

In this section, we conduct a series of ablation studies on the NNLQP datasets to investigate the impact of various modifications. We conduct comparative experiments under the different distributions of training and testing data, and the SqueezeNet family is selected as the test domain. As shown in

Table 5: Accuracy prediction results on NAS-Bench-101 (Ying et al., 2019) & NAS-Bench-201 (Dong & Yang, 2020) . We use different proportions of data as the training set and report Kendall's Tau on the whole datasets.

| Method | Publication | NAS-Bench-101 | | | NAS-Bench-201 | | |
|---|---|---|---|---|---|---|---|
| | | 0.04% (172) | 0.1% (424) | 1% (4326) | 3% (469) | 5% (781) | 10% (1563) |
| NP (Wen et al., 2020) | ECCV 2020 | 0.545 | 0.679 | 0.769 | 0.584 | 0.634 | 0.646 |
| Graphormer (Ying et al., 2021) | NeurIPS 2021 | 0.580 | 0.611 | 0.797 | 0.680 | 0.719 | 0.776 |
| TNASP (Lu et al., 2021) | NeurIPS 2021 | 0.669 | 0.705 | 0.820 | 0.640 | 0.689 | 0.724 |
| NAR-Former (Yi et al., 2023a) | CVPR 2023 | 0.653 | 0.765 | 0.871 | 0.790 | 0.849 | 0.901 |
| PINAT (Lu et al., 2023) | AAAI 2024 | 0.715 | 0.772 | 0.846 | 0.706 | 0.761 | 0.784 |
| NAR-Former V2 (Yi et al., 2023b) | NeurIPS 2023 | 0.704 | 0.773 | 0.861 | 0.846 | 0.874 | 0.888 |
| NN-Former (Xu et al., 2025) | CVPR 2025 | **0.765** | 0.809 | 0.877 | 0.860 | 0.879 | 0.890 |
| Ours | - | 0.762 | **0.809** | **0.880** | **0.864** | **0.881** | **0.892** |

Table 4, we obtain the following two conclusions: **(1) The proposed dynamic graph self-attention (DGSA) substantially improves the representation quality of neural architectures.** Row 1 presents the primary baseline, NN-Former, which uses its original position embedding and the ASMA attention mechanism proposed in that work. Replacing DGSA with global attention (Row 2) leads to a clear accuracy drop, confirming the effectiveness of our proposed graph attention. Row 4, which applies DGSA with position embedding, already improves over the baseline by reducing MAPE by 2.99% and increasing Acc(10%) by 3.25%, demonstrating that DGSA alone yields significant gains. Row 3 indicates that fixing the dynamic weights in Equation (4) results in a 5.05% decrease in Acc(10%) compared to Row 6, highlighting the benefit of adaptive adjacency selection. **(2) Language embedding provides richer and deeper semantic modeling capabilities for the model.** Replacing language embedding with position embedding (Row 4) reduces accuracy by 2.00%, while discarding pretraining (Row 5) causes a 16.65% decline. These results emphasize that both DGSA and pretrained language embeddings are critical to the effectiveness and generalization of our model.

### 4.4 ACCURACY PREDICTION

To further evaluate the generalization capability of our approach, we conduct accuracy prediction experiments on NAS-Bench-101 and NAS-Bench-201, shown in Table 5. While LeDG-Former also achieves strong performance on NAS-Bench-101 and NAS-Bench-201, the improvement over the state-of-the-art NN-Former is relatively marginal compared to the substantial gains observed on the NNLQP benchmark. We attribute this to two main factors: **First**, our dynamic graph-based modeling is particularly effective for architectures with deep and complex topologies, whereas cell-based search spaces typically contain shallow architectures with only 5 to 7 operations, limiting the richness of structural information that can be exploited. **Second**, the relatively small number of unique architectures and training samples in these benchmarks may lead to saturated prediction performance, reducing the observable performance gap. Despite this, the consistent results across diverse settings further demonstrate the robustness of LeDG-Former.

## 5 CONCLUSION

In this paper, we propose LeDG-Former, a novel neural architecture representation learning framework that synergistically integrates hardware-aware language embedding and dynamic graph-based transformer modeling. Our framework addresses the limitations of existing methods by incorporating hardware attributes and employing dynamic adjacency structures to effectively capture fine-grained structural differences among computational nodes. By projecting both neural architectures and hardware specifications into a unified semantic embedding space through language-model tokenization, LeDG-Former achieves the first successful zero-shot latency prediction across diverse hardware platforms on the NNLQP dataset. Comprehensive experiments further demonstrate that our approach surpasses existing state-of-the-art methods across multiple architecture-property prediction benchmarks, including NAS-Bench-101 and NAS-Bench-201. These findings highlight the importance of hardware-awareness and dynamic topology modeling for deployability-aware neural architecture representation. However, existing cross-hardware latency datasets cover limited hardware and architectural diversity, hindering robust evaluation under domain shifts. Future work should develop more diverse benchmarks to support comprehensive and realistic assessments.

ETHICS STATEMENT

This work adheres to the ICLR Code of Ethics. Our research does not involve human subjects, sensitive personal data, or potentially harmful applications. All datasets used in this paper are publicly available and widely used in the research community, and no ethical concerns arise from their usage. We confirm that there are no conflicts of interest or ethical issues related to the methods or experiments presented.

REPRODUCIBILITY STATEMENT

We have taken measures to ensure the reproducibility of our results. The details of the model architecture, training setup, datasets, and evaluation protocols are clearly described in the main paper. Additional implementation details, hyperparameters, and ablation studies are provided in the appendix and supplementary materials. The datasets used are publicly available, and we will release the source code and scripts for data processing and training in the camera-ready version.

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

## A    SUPPLEMENTARY MATERIALS

### A.1    LLM USAGE STATEMENT

Large Language Models were only used for grammar refinement and language polishing of this manuscript.

### A.2    EXPERIMENTS DETAILS

In this section, we provide detailed descriptions of the datasets, experimental settings, and training configurations used in our study. For each task, we follow established protocols and apply consistent training strategies to ensure fair and reproducible comparisons.

For accuracy prediction, we evaluate on two cell-structured benchmarks: NAS-Bench-101 and NAS-Bench-201. NAS-Bench-101 contains 423,624 unique architectures. We use training subsets of 0.04%, 0.1% and 1% of the full dataset and evaluate on the entire test set. NAS-Bench-201 includes 15,625 cell candidates, and we conduct training on 3%, 5% and 10% subsets before evaluating on the full set. For latency prediction, we use the NNLQP dataset, which consists of 20,000 deep-learning architectures annotated with latency measurements across multiple hardware platforms. The latency measurements of these architectures on a single hardware platform are used in the "Latency Prediction on NNLQP" experiments in the main text and are referred to as the "unseen" dataset. Additionally, a subset of these architectures is measured on multiple heterogeneous hardware platforms, forming the "multi_platform" datasets used in the "Hardware-Aware Zero-Shot" experiments described in the main text.

All experiments are trained using the Adam optimizer with a linear learning rate schedule: the learning rate warms up to 0.001 during the first 10% of training steps and then linearly decays to zero. Training is conducted on a machine equipped with an NVIDIA GeForce RTX 3090 GPU. For latency prediction tasks, we conduct 10 independent experiments under each configuration and report the average results to ensure stable evaluation.

### A.3    ABLATION STUDIES

To further validate the effectiveness of our proposed components and provide deeper insight into model behavior, we conduct extensive ablation studies. These results not only support the effectiveness of our proposed components but also offer practical insights for future developments in neural architecture modeling and deployment-aware design.

**Dynamic Graph Selection**    Firstly, to validate that our proposed Dynamic Graph Self-Attention mechanism effectively and adaptively selects different adjacency relationships, we conducted experiments on samples with varying topological structures. Specifically, we printed the dynamically selected attention mask weights for different nodes within these structures to visually illustrate the effectiveness of our dynamic selection mechanism.

As shown in Table 6, we selected one structural sample each from Mnasnet, Mobilenetv2, and Resnet18 architectures, verifying that our method generalizes broadly across different types of neural network structures. In particular, we chose sample 02001 from Mnasnet, sample 04001 from Mobilenetv2, and sample 06001 from Resnet18, these identifiers are original indices from the publicly available "unseen" subset of the NNLQP dataset. To further demonstrate our mechanism's capability to dynamically attend to distinct adjacency relationships based on the topological position of nodes, we analyzed nodes located at different positions within the aforementioned structures.

In the Resnet18 sample, we selected node_0 and node_6, where node_0 is the first operational node immediately after the input (preceding the first residual connection), and node_6 is the target node of the first residual connection. As presented in Table 6, node_0, being the initial node, exhibits attention weights primarily biased toward downstream child nodes. In contrast, the attention weights for node_6 distinctly shift focus toward the grandfather-level adjacency relationships, accurately reflecting the topological characteristics inherent in residual connection structures.

For the Mnasnet sample, we selected node_7 and node_15, where node_7 is situated before the first residual connection, and node_15 is the node targeted by this residual connection. Compared to

Table 6: Ablation study of the dynamic selection of attention masks in Dynamic Graph Self-Attention. The table reports attention mask weights assigned to $M_{Grandfather}$, $M_{Father}$, and $M_{Son}$ for each selected node.

| Model Type | Node No. | $M_{Grandfather}$ | $M_{Father}$ | $M_{Son}$ |
|---|---|---|---|---|
| Resnet18 | node_0 | 0.3304 | 0.3312 | 0.3383 |
| | node_6 | 0.3328 | 0.3327 | 0.3344 |
| Mnasnet | node_7 | 0.3318 | 0.3336 | 0.3345 |
| | node_15 | 0.3338 | 0.3324 | 0.3337 |
| Mobilenetv2 | node_12 | 0.3325 | 0.3332 | 0.3342 |
| | node_16 | 0.3331 | 0.3332 | 0.3337 |

Table 7: Ablation study of hardware embedding token. Setting on Zero-shot latency prediction on reorganized NNLQP "multi_platform" datasets, Nvidia Tesla P4 → Nvidia Tesla T4.

| Metric | NN-Former | w/o hardware embedding | **Ours** |
|---|---|---|---|
| Average MAPE ↓ | 54.13 | 41.40 | 19.06 |
| Average Acc(10%) ↑ | 7.91 | 8.15 | 34.62 |

node_0 in Resnet18, which has only a single preceding node, node_7 possesses multiple predecessor nodes; accordingly, its dynamically selected attention weights exhibit a pronounced focus on predecessor adjacency relationships (both father and grandfather). This result underscores the effectiveness of our mechanism in dynamically attending to adjacency relationships according to node-specific topological context.

Similarly, for the Mobilenetv2 sample, we analyzed nodes node_12 and node_16, node_12 is located within the first residual connection block, while node_16 follows this residual connection. The dynamically selected attention weights for these two nodes similarly conform to the logical expectations based on their respective topological positions.

Collectively, these experimental results substantiate that our proposed Dynamic Graph Self-Attention mechanism genuinely and effectively performs dynamic adjacency selection, confirming the importance and validity of dynamically modeling distinct adjacency information for nodes located at varying topological positions during neural architecture representation learning.

**Hardware Embedding**  To further validate the contribution of the hardware embedding token, we performed an additional ablation study under the zero-shot transfer setting from Nvidia Tesla P4 → Nvidia Tesla T4. As shown in Table 7, removing the hardware embedding led to a severe degradation in performance. In particular, the Average MAPE increased from 19.06 to 41.40, while the Acc(10%) dropped sharply from 34.62% to 8.15%, corresponding to a 26.47% decrease. This substantial performance gap highlights that hardware-aware representations are crucial for accurate and generalizable modeling across heterogeneous platforms. These results further reinforce our claim that the proposed hardware embedding token is indispensable for achieving robust cross-platform latency prediction.

**Language Models Adopted**  In the language embedding stage of LeDG-Former, the language model is proposed as a replaceable module whose performance is closely related to the overall predictive capability of the method. In our main experiments, we use the smallest BERT ("bert-base-uncased"), which already achieves strong results as reported in the main text. Nevertheless, to further examine the influence of language models, we conducted an ablation study on the latency prediction task using the NNLQP dataset.

As shown in Table 8, in Rows 1, we first randomly initialized the parameters of the originally employed BERT model to perform language embedding. By comparing Rows 1 with Rows 2, we observe a significant drop in performance, demonstrating the critical importance of semantic modeling provided by pretrained language models for this task. Moreover, in Rows 3, we replaced the original

Table 8: Ablation study on the impact of different language models for latency prediction on the NNLQP dataset.

| Rows | Language Models | MAPE↓ | Acc(10%)↑ |
|------|-----------------|-------|-----------|
| 1 | bert-base-uncased randomly initialized | 8.25 | 66.45 |
| 2 | bert-base-uncased | 2.64 | 97.94 |
| 3 | bert-large-uncased | 2.57 | 98.10 |

model with a larger-scale language model ("bert-large-uncased"), resulting in a decrease in MAPE by 0.07 and an increase in Acc(10%) by 0.16%. This result further confirms our hypothesis that richer language embeddings contribute to better generalization performance. In theory, employing even larger-scale language models might further enhance performance, which we regard as a promising direction for future research.

