# OpenReview forum: "Language Embedding Meets Dynamic Graph: A New Exploration for Neural Architecture Representation Learning"
_ICLR.cc/2026/Conference — ICLR 2026 Conference Withdrawn Submission_

### Official Review · Reviewer_x8ou · 2025-10-27

**Soundness:** 3
**Presentation:** 2
**Contribution:** 3
**Rating:** 6
**Confidence:** 4

**Summary:**

The paper addresses the insufficient information usage problem in designing neural architecture representation learning methods. It identifies two key limitations, which are the overlooked hardware information and the heterogeneity of the nodes in the architecture graph. To tackle these issues, the authors propose a novel architecture representation learning method that incorporates hardware information and utilizes a language template representation to capture the heterogeneity of nodes. Three different masks are applied to process the attention information. The proposed method is evaluated on several NAS benchmarks, demonstrating its effectiveness compared to existing baselines.

**Strengths:**

1. The paper addresses the insufficient information usage problem in designing neural architecture representation learning methods. It identifies two key limitations, which are the overlooked hardware information and the heterogenety of the nodes in the architecture graph, which are well motivated.

**Weaknesses:**

1. Although the language template representation part is novel in the NAS domain, it is still not a novel solution. In areas like graph foundation model or graph learning, related techniques that transform graphs into tokens by using LLMs have already been explored.
2. There lacks an in-depth investigation of the proposed problem. The positive correlation between the hardware information and the resulting latency is trivial to find. It is better to provide more insights than only showing the performance improvements.

**Questions:**

1. What are the specific reasons for using three masks (son, father, grandfather)? What about using more or less masks?
2. Table 1 is not straightforward to read. Please clarify the meaning of baselines at the beginning of the section.

---

### Official Review · Reviewer_wbrn · 2025-10-30

**Soundness:** 3
**Presentation:** 1
**Contribution:** 3
**Rating:** 2
**Confidence:** 5

**Summary:**

Paper is interested in proposing an ML model that receives a neural architecture as input, and outputs information about it, including predicting latency of running the input neural architecture on specific hardware (useful for compiler/program optimization, without having to compile and run every program); or predicting the accuracy of the input architecture on classification tasks (useful for increasing accuracy, without having to train every hyperparameter configuration). The paper produces estimates that are better than other SoTA methods.

I think the paper has a lot of good contributions, however, I will vote for rejection due to limited novelty (model is straightforward combination of others) and also inaccurate advertising ("dynamic" graph & generalizing to unseen opcodes). However, if paper revises this (e.g., removes word dynamic from everywhere, and admits that the novelty is in the application and not in the model invention), then I could upgrade my rating.

**Strengths:**

* The paper sets state-of-the-art for a couple of problems where the input is some neural network architecture. This has broad use-cases in "efficiency" [running faster] and in neural architecture search [finding more accurate networks, without going through the training process].

* The paper introduces a GNN architecture, which consists of combining known techniques

* Paper combines LLMs with GNNs. This combination is an active area of research.

**Weaknesses:**

## Major weaknesses

* The model is attention model on multiple adjacency matrices, i.e., a straightforward combination of (MixHop, NGCN, or alike) with graph transformer archiectures.

* The word dynamic should be completely eliminated from everywhere: title, section headers, abstract, and main text. The paper does *not* deal with dynamic graphs. It deals with static graphs. They only combine different adjacency hops from a fixed graph (like MixHop, etc). Dynamic implies that the neural network architecture is changing with time (which is the case, for "dynamic neural networks", which are not studied at all in this paper).

* The computational complexity of the method (eq 6-8) seems to be quadratic -- due to term $Q K^\top$. This is actually bad news, especially because graphs of neural networks can contain tens-of-thousands, up-to a couple of million nodes.

* Last paragraph of Intro claims that prior methods are limited to single hardware optimization. This is false. While I am not an expert in the field, I happened to attend the oral of https://openreview.net/pdf?id=bpS4vaOg7q which also does training & inference on multiple hardwares.

* "The specialized language templates" line 101 defeats the motivation to handle "unseen node types" [line 188]. Can the authors show experiments on generalizing to new op-codes? In any case, I personally dont think this is a big issue because I would assume the op-codes in neural architectures are merely a handful (less than 300?): conv, add, relu, einsum, etc.

* Please mention the sizes of your matrices and vectors (parameters and inputs/outputs). For instance, the dimensionality of $W_1, W_2, W_3$ (of Eq.4) is not obvious. This is especially confusing because softmax usually returns one object (not 3). The model is the most important piece (IMO) and being exact is important.

* The representation $f^r_{node}$ is not used after definition. Perhaps math bug?

* The related section does not serve its purpose and can be cut by at least 70%. Why not just focus on GNNs applied on Neural Architectures?

## Minor Weaknesses

* "model deployment" (last words of 4th line of paper) is very out of place. I think the paper meant to say "inference"?

* In many paragraphs, the paper writes "idea1, idea2, idea3 (citation1, citation2, citation3)". It is better to properly distribute as "idea1 (citation1), idea2 (citation2), ...".

**Questions:**

* Did you actually do the quadratic computation of $Q K^\top$? If not, please detail in the paper how.

* Please address other questions from the weaknesses

---

### Official Review · Reviewer_RKFN · 2025-11-02

**Soundness:** 2
**Presentation:** 3
**Contribution:** 2
**Rating:** 4
**Confidence:** 3

**Summary:**

This paper focuses on neural architecture representation learning, which typically represents computational operations as nodes and data flow as edges. Although transformer–GNN-style models have successfully captured both local and global information in graphs, current designs still overlook hardware attribute specifications and positional variations among nodes, as well as their distinct neighborhood attention patterns. To address these issues, this work integrates large language models (LLMs) with dynamic graphs to better capture hardware attributes and improve flexibility.

**Strengths:**

1. The proposed framework effectively integrates neural architecture and hardware platform representations, advancing existing approaches.
2. The proposed dynamic graph self-attention mechanism reduces the computational cost of full attention while maintaining flexibility for specific local structures.
3. The experimental results appear strong and convincing.

**Weaknesses:**

1. Is the predefined template sufficiently informative for any neural architecture design? The manually defined template may be limited by the user’s knowledge. Could an automated approach be developed to extract more aligned concepts from both the neural architecture and the hardware platform? Furthermore, how are the hardware platform descriptions obtained? Their quality may significantly affect overall performance.
2. If the attended nodes are predefined by nearly two-hop parents, how can the model still benefit from the global information flow that the original graph transformer provides?

**Questions:**

My main concern is W1 as described alove. Besides, there is a typo: “Attributes Predcting” in Figure 1.

---

### Author Response · Authors · 2025-11-14

Dear Area Chair and Reviewers,

We sincerely thank the Area Chair and all reviewers for their time, effort, and constructive feedback.

To briefly clarify a key misunderstanding (especially by Reviewer wbrn) about our novelty: LeDG-Former is a hardware-aware neural architecture representation framework that jointly encodes architectures and hardware platforms via language embeddings and a dynamic graph transformer. By tokenizing both network structures and hardware specifications into a shared LLM-based semantic space, and using a dynamic graph module that learns adaptive relations instead of fixed adjacency matrices, LeDG-Former yields more expressive and hardware-aware representations. Experiments on NNLQP and NAS-Bench-101/201 show that LeDG-Former sets new state-of-the-art results and, for the first time, achieves zero-shot latency prediction on unseen hardware platforms.

Once again, we are grateful for the reviews, which will greatly help us to strengthen and clarify this line of work.

Best regards,

The Authors

---

### Note · Authors · 2025-11-14

I have read and agree with the venue's withdrawal policy on behalf of myself and my co-authors.